# Sterilization Effects on Liposomes with Varying Lipid Chains

**DOI:** 10.3390/nano15191478

**Published:** 2025-09-27

**Authors:** Sarocha Cherdchom, Krit Pongpirul, Natchanon Rimsueb, Prompong Pienpinijtham, Amornpun Sereemaspun

**Affiliations:** 1Center of Excellence in Preventive and Integrative Medicine (CE-PIM), Faculty of Medicine, Chulalongkorn University, Bangkok 10330, Thailand; sarocha.cherdchom@gmail.com; 2Department of Preventive and Social Medicine, School of Global Health, Faculty of Medicine, Chulalongkorn University, Bangkok 10330, Thailand; 3Bumrungrad International Hospital, Bangkok 10110, Thailand; 4Department of Infection Biology & Microbiomes, Faculty of Health and Life Sciences, University of Liverpool, Liverpool L69 7ZX, UK; 5National Nanotechnology Center (NANOTEC), National Science and Technology Development Agency (NSTDA), Ministry of Higher Education, Science, Research and Innovation, 111 Thailand Science Park, Phahonyothin Road, Klong Nueng, Klong Luang, Pathum Thani 12120, Thailand; natchanon.rim@nanotec.or.th; 6Department of Chemistry, Faculty of Science, Chulalongkorn University, Bangkok 10330, Thailand; prompong.p@chula.ac.th; 7Center of Excellence in Nanomedicine, Department of Anatomy, Faculty of Medicine, Chulalongkorn University, 1873 Rama 4 Road, Pathumwan, Bangkok 10330, Thailand; amornpun.s@gmail.com

**Keywords:** liposome, sterilization, DPPC and DSPC, physicochemical properties, lipid nanoparticles

## Abstract

Liposomes, nanoscale vesicles with distinct structural and functional properties, are widely utilized in drug delivery due to their biocompatibility and ability to encapsulate diverse therapeutic agents. Effective sterilization is essential to ensure the safety and efficacy of liposomal formulations in biomedical applications, yet its impact on liposome integrity and functionality remains inadequately studied. This work systematically evaluates the effects of three sterilization methods: autoclaving, UV radiation, and filtration—on liposomes composed of dipalmitoylphosphatidylcholine (DPPC) and distearoylphosphatidylcholine (DSPC), two phospholipids differing in lipid chain length. Sterilization altered liposome properties in a lipid chain length-dependent manner, affecting particle size, zeta potential, and phospholipid content. Filtration caused significant hydrocarbon loss, confirmed by Fourier-transform infrared spectroscopy (FTIR) and Raman spectroscopy, and led to a higher reduction in phospholipid content in DPPC liposomes compared to DSPC liposomes. Biological evaluations showed that autoclaved and UV-irradiated DPPC liposomes exhibited higher cytotoxic and lower stability than their DSPC counterparts. While autoclaving and UV irradiation resulted in minimal chemical alterations, both methods significantly influenced biological properties. Filtration, although less disruptive to biocompatibility, also reduced key liposomal integrity and efficacy. This study underscores the critical importance of post-sterilization evaluation to optimize liposomal formulations for clinical and biomedical use.

## 1. Introduction

Nanomaterials, engineered at the nanoscale, play a pivotal role in various nanotechnology applications. Among these, liposomes have emerged as one of the most extensively studied and utilized systems in drug delivery [1]. These versatile and biocompatible nanoscale structures are composed of lipid bilayers that mimic cell membranes, encapsulating an aqueous core [2]. Owing to their unique properties—such as high biocompatibility, customizable surface chemistry, and the ability to encapsulate both hydrophilic and hydrophobic molecules—liposomes are promising vehicles for drug delivery, gene therapy, and vaccine development [3,4]. Beyond biomedical applications, liposomes are widely explored in functional food, nutraceuticals [5], and cosmetics [6] for their ability to enhance health benefits and improve product efficacy. Among the available fabrication approaches, the thin-film hydration method [7] is widely adopted for its simplicity, scalability, and ability to produce liposomes with defined characteristics suitable for a range of applications [8]. This method allows for efficient drugs and therapeutic agent incorporation, enhancing delivery efficiency and therapeutic potential [8,9,10,11,12].

Despite their numerous advantages, liposomes face challenges, particularly in sterilization, a critical step for ensuring safety and efficacy in biomedical applications. The synthesis of liposomes typically involves the hydration of lipid films, followed by sonication or extrusion to achieve uniform vesicles with the desired size and composition. However, sterilization processes can alter the physicochemical properties of liposomes, potentially affecting their stability, drug release kinetics, and biological activity.

Effective sterilization is essential to eliminate microbial contaminants, which pose significant risks to patient safety [13,14]. Sterilization can be performed through physical or chemical methods, each with inherent advantages and limitations [15]. Physical sterilization techniques include autoclaving, ultraviolet (UV) irradiation, and filtration. Autoclaving employs high-pressure steam to inactivate microbes, while UV irradiation uses ultraviolet light to disrupt microbial DNA. Filtration relies on sterilizing filters with defined pore sizes to physically remove contaminants. In contrast, chemical methods, such as ethylene oxide (EtO) gas or hydrogen peroxide vapor, use chemical agents to achieve microbial inactivation [14,16]. Physical methods are often preferred for nanoparticles and liposomes due to their simplicity, efficiency, and compatibility with delicate nanostructures. However, these processes can introduce changes to the liposomes’ structural and functional properties, necessitating careful evaluation [17]. The choice of sterilization method significantly influences the physicochemical properties of liposomes, particularly in formulations composed of phospholipids with varying fatty acyl chain lengths and phase transition temperatures, such as DPPC and DSPC [18,19,20]. DPPC, with its two saturated palmitoyl (16:0) fatty acid chains, has a melting temperature (T_m_) of approximately 41 °C [21], making it more fluid and permeable at physiological temperatures—an advantage for controlled drug release. Conversely, DSPC, with stearoyl (18:0) fatty acid chains, has a higher T_m_ of around 54 °C, resulting in more rigid and stable liposomes [22]. The phospholipid composition significantly impacts liposome stability and drug release kinetics [19], underscoring the importance of selecting appropriate sterilization techniques that preserve liposomal functionality.

Despite the importance of sterilization, there is a lack of systematic studies on how various sterilization techniques affect the properties of liposomes. Understanding these effects is critical for optimizing liposomal formulations and ensuring consistent product quality.

This study represents the pioneering effort to assess and compare the impact of three physical sterilization methods—autoclaving, UV irradiation, and filtration—on the physicochemical and biological properties of liposomes. Liposomes synthesized from dipalmitoylphosphatidylcholine (DPPC) and distearoylphosphatidylcholine (DSPC), two commonly used phospholipids with distinct chain lengths [23], were evaluated. Key characteristics, including particle size, polydispersity index (PDI), surface charge, and biological compatibility, were analyzed in sterilized and non-sterilized formulations. Investigating these sterilization effects is essential for developing sterilization strategies that preserve liposome stability and biocompatibility, thereby advancing liposome-based technologies for innovative drug delivery and disease treatment.

## 2. Materials and Methods

### 2.1. Materials

1,2-dipalmitoyl-sn-glycero-3-phosphocholine (DPPC) and 1,2-distearoyl-sn-glycero-3-phosphocholine (DSPC) were obtained from Avanti Polar Lipids (Darmstadt, Germany). Cholesterol was sourced from Avanti Polar Lipids (Birmingham, AL, USA), and didecyldimethylammonium bromide (DDAB) was obtained from Sigma–Aldrich Chemical Company (St. Louis, MO, USA). Phosphate-buffered saline (PBS) at pH 7.4 (containing 137 mM NaCl, 2.7 mM KCl, 10 mM Na_2_HPO_4_; 2 mM KH_2_PO_4_) and ethanol were purchased from Merck (Darmstadt, Germany).

### 2.2. Preparation of Liposome

Liposomes were prepared using the conventional thin film hydration method [24]. Lipids (DPPC or DSPC) and cholesterol, along with DDAB, were dissolved in ethanol at a molar ratio of 80:15:5. The solvent was removed using rotary evaporation at 25 °C under vacuum to form a thin lipid film (approximately 8 h). The dried lipid film was rehydrated with distilled water and sonicated at 25 °C for 5 min. The obtained liposomes were stored in a refrigerator under dark conditions. The total lipid concentration was 3 mg/mL for DPPC and 2.5 mg/mL for DSPC.

### 2.3. Liposomes Sterilization Methods

DPPC and DSPC liposomes were sterilized using three methods: autoclaving (A), UV irradiation (U), and filtration (F).

Autoclaving: Samples were sterilized in a HIRAYAMA autoclave (Hirayama Manufacturing. Corp., Saitama, Japan) at 121 °C for 15 min.

UV irradiation: Liposomes were transferred to 1.5 mL polypropylene microcentrifuge tubes (Axygen, VWR, Mississauga, ON, Canada) and exposed to UV light at 254 nm (15 W) for 15 min at room temperature. The tubes were positioned vertically with lids open, approximately 24 inches below the UV lamp.

Filtration: Samples were passed through an Acrodisc syringe filter (0.2 µm pore size, polyethersulfone (PES) membrane, Pall Medical, Fribourg, Switzerland) at room temperature using manual syringe pressure.

After sterilization, all samples were cooled to room temperature, stored in a closed container, and refrigerated at 4 °C. Non-sterilized liposomes (C) served as controls.

### 2.4. Liposome Characterization

#### 2.4.1. Particle Size, Polydispersity, and Zeta Potential Determination

The particle size, polydispersity index (PDI), and zeta potential were measured using a ZS90 particle analyzer (Malvern Instruments, Malvern, UK). Liposome samples (1 mL) were diluted with deionized distilled water. Particle size and PDI were determined using the dynamic light scattering technique at 25 °C, while zeta potential was measured based on electrophoretic mobility using DTS software version 1.41 (Malvern, UK). Results are presented as mean ± standard deviation (SD).

#### 2.4.2. pH Measurements

The pH of liposome samples was measured before and after sterilization using pH meter (Mettler Toledo InLab^®^ Micro Pro-ISM Electrode with a SevenDirect SD20, Columbus, OH, USA).

#### 2.4.3. Phospholipid Measurement

Phospholipid content in DPPC and DSPC liposomes was quantified using the Choline Oxidase DAOS method (Fujifilm Wako), following the manufacturer’s instructions. The manufacturer’s protocol is available at: https://labchem-wako.fujifilm.com/us/product_data/docs/04093472_doc01.pdf (accessed on 19 August 2025).

#### 2.4.4. Fourier-Transform Infrared (FTIR) Spectroscopy

FTIR spectra were collected using a Thermo Fisher Scientific Nicolet iS5 FT-IR Spectrometer (Waltham, MA, USA) in attenuated total reflection (ATR) mode with an iD7 accessory. The deuterated triglycine sulphate (DTGS) detector operated with 16 scans at a spectral resolution of 4 cm^−1^. The spectra were normalized without additional modification.

#### 2.4.5. Raman Spectroscopy (RS)

Raman analysis was performed using a Horiba LabRAM HR Evolution Confocal Raman Microscope (Horiba Scientific, Kyoto, Japan) with a 532 nm laser source. Samples were analyzed as powders with acquisition times of 25 s, eight accumulations, and a grating at 1800 grooves/mm. The slit width was 200 µm, and the hole size was 400 µm. Raman spectra were analyzed using LabSpec6 (Horiba Scientific, Kyoto, Japan) and OriginLab2022 software.

### 2.5. Biological Evaluation

#### 2.5.1. Cell Cultures

HaCaT (human epidermal keratinocyte), HepG2 (human hepatocellular liver carcinoma), and HK-2 (human renal proximal tubular epithelial cell) cells were cultured in Dulbecco’s Modified Eagle’s Medium (DMEM, Corning, Manassas, VA, USA), supplemented with 10% Gibco Fetal Bovine Serum (Thermo Fisher Scientific, Waltham, MA, USA) and 1% Gibco Penicillin-Streptomycin (Thermo Fisher Scientific, Waltham, MA, USA). Cells were maintained at 37 °C in a humidified incubator with 5% CO_2_, with media refreshed every 48 h. Cells were passaged at 70–80% confluence and sub-cultured for cytotoxicity testing.

#### 2.5.2. Cytotoxicity Evaluation

The cytotoxicity of sterilized and non-sterilized liposomes was assessed using PrestoBlueTM reagent (Invitrogen, Waltham, MA, USA). Cells were seeded into 96-well plates at a density of 1 × 10^4^ cells/well and incubated for 24 h. After adherence, cells were treated with different liposome concentrations for 24 and 48 h. Detailed concentration-dependent viability data for DPPC and DSPC liposomes before and after sterilization are provided in Appendix A. PrestoBlue reagent (10 µL) was added to each well, followed by a 30 min incubation. Fluorescence was measured at 560 nm excitation and590 nm emission using a microplate reader (Thermo, Varioskan Flash, Birmingham, UK).

### 2.6. Statistical Analysis

All experiments were performed in triplicate, and results are expressed as the mean ± standard deviation (*n* = 3). Statistical analysis was conducted using one-way ANOVA and Tukey’s post-test with GraphPad Prism 9 (GraphPad Software Inc., San Diego, CA, USA). Differences were considered significant at *p* < 0.05.

## 3. Results

### 3.1. Physical Appearance and Physicochemical Characteristics

Following sterilization via autoclaving, filtration, and UV irradiation, both DPPC and DSPC liposomes exhibited no visible changes in physical appearance compared to non-sterilized controls. The initial particle sizes of DPPC and DSPC liposomes were approximately 100 nm and 120 nm, respectively. After sterilization, autoclaving, and filtration significantly reduced particle sizes for both DPPC and DSPC liposomes compared to the control group (Figure 1, Table 1). Despite these reductions, the polydispersity index (PDI) remained stable across all conditions, indicating maintained size uniformity. However, filtration led to a significant decrease in zeta potential for both types of liposomes, suggesting alterations in surface charge properties.

### 3.2. pH Values Before and After Sterilization

Initial pH measurements indicated that DPPC liposomes had lower pH values than DSPC liposomes, likely due to differences in lipid composition. Post-sterilization, distinct pH alterations were observed across treatments. Autoclaving and UV irradiation resulted in a pH, indicating acidification, while filtration led to an increase, suggesting a shift toward alkalinity. Notably, DSPC liposomes exhibited a significant pH reduction after autoclaving and filtration, whereas UV irradiation had a negligible impact on pH levels (Table 2).

### 3.3. Influence of Sterilization on Chemical Characteristics of Liposomes

Fourier-transform infrared (FTIR) spectroscopy confirmed the characteristic peaks of both DPPC and DSPC liposomes across all sterilization methods (Figure 2). These peaks, corresponding to various functional groups [25], are detailed in Table 3.

Liposomes subjected to autoclaving and UV irradiation maintained strong, well-defined spectral bands, suggesting that these treatments preserved molecular organization and lipid packing. However, filtration-sterilized liposomes exhibited significant spectral alteration, including reductions in specific peaks at 721, ~1376, 1467, ~1735, 2849, ~2915, and 2955 cm^–1^ and general weakening of peaks at ~825, 969, ~1062, 1088, ~1162, and ~1140 cm^–1^). These changes suggest that filtration may lead to structural modifications, particularly in the hydrocarbon chains of the lipid bilayer.

A detailed comparison of band positions, relative intensities, and linewidths (FWHM) for hallmark FTIR modes—phosphate ν(PO_2−_) (~1088–1090 and 1236–1242 cm^−1^), ester carbonyl ν(C=O) (~1734–1736 cm^−1^), methylene ν_s_ (CH_2_) (~2849 cm^−1^) and ν_as_ (CH_2_) (~2915–2916 cm^−1^), together with CH_2_ scissoring/rocking (~1467/721 cm^−1^)—revealed that autoclaving and UV produced only minor shifts (≤1–2 cm^−1^), negligible intensity changes, and no new absorptions. In contrast, filtration caused consistent attenuation of hydrocarbon-chain vibrations, weakening of headgroup-associated peaks, and modest broadening of CH_2_ stretching bands, all of which suggest increased acyl-chain disorder and partial headgroup reorganization rather than the emergence of new chemical functionalities. Subtle lipid-dependent differences were also observed: DSPC spectra showed narrower baseline linewidths and slightly reduced broadening after filtration compared with DPPC, consistent with the inherently greater stability of longer acyl chains.

### 3.4. Comparison of Raman Spectra

Raman spectroscopy was used to further investigate the structural integrity of liposomes post-sterilization (Figure 3). Both DPPC and DSPC liposomes exhibited characteristic peaks indicative of cholesterol, particularly in the 2800–2950 cm^−1^ range, corresponding to the symmetric stretching of CH_2_ and CH_3_ groups. Additionally, the C-C steroid ring vibration in the cholesterol structure was identified at 1442 cm^−1^ [26].

The 700–750 cm^−1^ peak, associated with the C-N bond stretching vibration of the choline headgroup, was observed in both DPPC and DSPC liposomes [27]. This peak remained stable after autoclaving and UV irradiation, suggesting that these sterilization methods preserved the structural integrity of the choline group. However, the 700–750 cm^−1^ peak was significantly reduced following filtration, indicating that this process may have disrupted the choline environment or caused structural modification in the lipid bilayer.

Further analysis of the 2800–2900 cm^−1^ region, attributed to the symmetrical and asymmetrical stretching vibrations of C-H bonds in methylene and terminal methyl groups, confirmed that autoclaving and UV irradiation did not significantly alter the lipid structure. In contrast, filtration resulted in the loss of the 2950 cm^−1^ peak, suggesting potential disruption of C-H and C-N bond vibrations within the liposomal bilayer. These spectral changes indicate that filtration may lead to lipid reorganization or selective removal of certain lipid components.

These findings highlight the differential impact of sterilization methods on liposome structure, with autoclaving and UV irradiation maintaining molecular integrity, whereas filtration may induce subtle but significant modifications in lipid organization.

### 3.5. Phospholipid Measurement

Quantitative analysis of phospholipid content revealed notable differences among sterilization methods (Appendix A). DPPC liposomes exhibited a significant reduction in phospholipid content following autoclaving and UV irradiation, consistent with phospholipid degradation or hydrolysis under harsh conditions. Filtration also led to phospholipid loss in both DPPC and DSPC liposomes, likely due to physical removal or entrapment of lipids within the filter matrix. In contrast, DSPC liposomes demonstrated resilience to phospholipid degradation during autoclaving and UV exposure, likely due to longer, more saturated hydrocarbon chains, which confer greater structural stability (Table 4).

### 3.6. Effects of Sterilization on Cell Viability

The in vitro cytotoxicity of liposomes before and after sterilization was assessed in various cell lines over 24 and 48 h across a concentration range of 0.01–0.5 mg/mL. Cytotoxicity results revealed that filtration-treated DSPC liposomes exhibited the highest IC_50_ values, indicating reduced toxicity, particularly in HaCaT and HepG2 cell lines at both time points (Figure 4). In contrast, no significant differences in viability were observed between sterilized and non-sterilized liposomes in HK-2 cells. Autoclaved and UV-irradiated DPPC liposomes showed lower IC_50_ values in HaCaT cells at 48 h, suggesting increased cytotoxicity. Similarly, DSPC liposomes exhibited reduced IC_50_ values following autoclaving in HepG2 cells, highlighting the differential effects of sterilization on liposomal biocompatibility.

## 4. Discussion

Liposomes are versatile nanoscale vesicles composed of lipid bilayers, widely used in biomedical applications due to their biocompatibility, ability to encapsulate both hydrophilic and hydrophobic molecules, and potential for targeted drug delivery [28]. Their applications extend across drug delivery, gene therapy, vaccine development, and diagnostic imaging [2,29,30,31], making them a critical component in modern medical and pharmaceutical advancements. Given their widespread use, maintaining structural integrity and functional properties during sterilization is essential for ensuring safety, efficacy, and stability in clinical and pharmaceutical settings. Various sterilization techniques, including thermal autoclaving, UV irradiation, and filtration, are commonly applied to liposomal formulations. Each method can influence physicochemical properties and biological interactions differently.

### 4.1. Impact of Sterilization on Physicochemical Properties

Our findings indicated that sterilization significantly influences the particle size, zeta potential, pH, chemical composition, phospholipid content, and cytotoxicity of DPPC and DSPC liposomes. Autoclaving and filtration significantly reduced particle size for both liposomal formulations, with autoclaving leading to greater size reduction than filtration. This effect is likely due to membrane compaction or fusion under high-pressure steam and elevated temperatures. Despite these changes, the polydispersity index (PDI) remained stable, indicating that liposome uniformity was largely preserved. Previous studies suggest that autoclaving can enhance monodispersity by disrupting larger aggregates [32,33]. In contrast, filtration removed larger aggregates while decreasing zeta potential, suggesting interactions between the lipid bilayer and filter material that altered surface charge properties [34].

These results underscore the importance of understanding how sterilization techniques impact liposome stability and charge dynamics, which can influence drug loading efficiency and cellular interactions. Additionally, sterilization affected pH, with autoclaving and UV irradiation leading to acidification, while filtration induced alkalization. These shifts may result from phospholipid degradation, hydrolysis, or ionic release, which can affect drug stability, liposome dispersion, and release properties [35]. Importantly, sterilization methods such as UV irradiation and autoclaving may induce phase-transition–dependent effects. Since DPPC and DSPC transition between gel and liquid-crystalline states at different temperatures, sterilization-induced heating or irradiation could differentially affect bilayer fluidity, packing, and permeability [36,37]. These phase-dependent changes may explain the observed variations in physicochemical properties, and their direct measurement represents a limitation of this study. Future work using thermal analyses, such as differential scanning calorimetry (DSC), would clarify the relationship between sterilization, lipid phase behavior, and liposome stability.

### 4.2. Chemical and Structural Alterations

Although filtration is often recommended for liposomal sterilization due to its mild processing conditions [13], our findings reveal potential limitations. Previous studies reported drug leakage and loss of liposomal components during filtration [38,39]. Our FTIR and Raman spectroscopy results confirm that filtration caused a loss of characteristic long-chain hydrocarbon peaks, particularly CH_2_ and CH_3_ groups, indicating removal or alteration of certain lipid components. This phenomenon could be attributed to mechanical shear stress, lipid adsorption onto the filter membrane, or selective lipid retention [40]. While some nanomaterials undergo filtration with minimal modifications [15,41], our results highlight the need for careful optimization to minimize lipid loss and maintain structural integrity. For clarity, a spectroscopy-based comparison of chemical and structural alterations in DPPC and DSPC liposomes across different sterilization methods is provided in Appendix A.

Phospholipids are essential to the structural integrity, functionality, and versatility of liposomes [42,43]. Phospholipid quantification further confirmed that DPPC liposomes exhibited greater phospholipid loss after autoclaving and UV irradiation, likely due to thermal degradation, hydrolysis, or oxidative damage [44]. In contrast, DSPC liposomes demonstrated greater stability, with no significant phospholipid degradation and phase transitions [45]. However, filtration resulted in the most substantial phospholipid loss in both DPPC and DSPC liposomes, supporting our spectroscopic findings that filtration leads to physical lipid removal within the filter matrix. Given these findings, future investigations should focus on systematically analyzing adsorption kinetics, filter membrane chemistries, and dose-dependent lipid retention in order to better elucidate the underlying mechanisms of lipid loss during filtration sterilization.

### 4.3. Effects on Cytotoxicity and Biocompatibility

Post-sterilization biological evaluation is critical for ensuring the safety and efficacy of liposomal formulations. Our results revealed notable differences in cell viability between DPPC and DSPC liposomes, highlighting the role of lipid composition in cytotoxicity. Autoclaved and UV-irradiated DPPC liposomes exhibited lower IC_50_ values, particularly in HaCaT and HepG2 cells, indicating increased cytotoxicity. This effect could stem from sterilization-induced lipid degradation, which might alter membrane stability and cellular uptake. Conversely, DSPC liposomes maintained higher stability and lower cytotoxicity, aligning with prior studies suggesting that DSPC-based liposomes exhibit superior biocompatibility [19,45].

Interestingly, filtration-treated liposomes demonstrated higher IC_50_ values, indicating reduced cytotoxicity. This trend may be attributed to the removal of cytotoxic lipid components, suggesting that filtration, while structurally disruptive, may also enhance biocompatibility. These findings underscore the importance of selecting sterilization techniques that preserve sterility while maintaining liposomal integrity and minimizing adverse biological effects.

## 5. Conclusions

This study highlights the differential effects of sterilization methods on DPPC and DSPC liposomes, emphasizing the importance of tailored sterilization protocols to maintain liposome stability, functionality, and biocompatibility.

Key findings included the following: (1) Autoclaving and filtration significantly reduced particle size, with autoclaving leading to greater compaction, (2) Filtration caused the greatest reduction in zeta potential and phospholipid content, indicating potential lipid loss, (3) UV irradiation had the mildest impact on chemical composition but still influenced cytotoxicity, (4) DPPC liposomes exhibited greater instability and cytotoxicity post-sterilization due to the removal of harmful lipid fractions.

These findings reinforce the need for strategic sterilization selection to balance sterility, liposomal integrity, and safety in biomedical applications. Future research should focus on developing optimized sterilization techniques that minimize structural and compositional alterations while ensuring sterility. This approach will enhance the safety and efficacy of liposomal drug delivery systems, paving the way for their broader clinical application.

## Figures and Tables

**Figure 1 nanomaterials-15-01478-f001:**
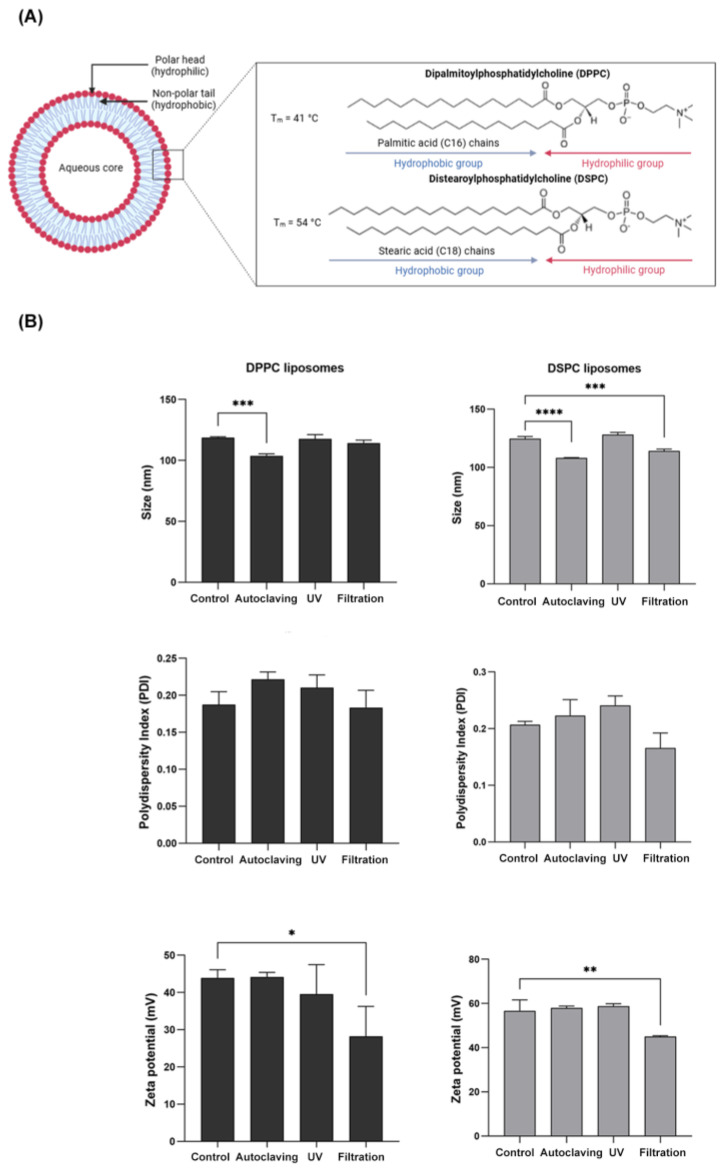
Structural representation and physicochemical characterization of DPPC and DSPC liposomes. (**A**) Schematic illustration of the structural differences between DPPC and DSPC lipid chains. (**B**) Particle size, PDI, and zeta potential of DPPC and DSPC liposomes following different sterilization treatments, including autoclaving, UV irradiation, and filtration, compared to non-sterilized control. Statistical significance: * *p* ≤ 0.05, ** *p* ≤ 0.01, *** *p* ≤ 0.001, **** *p* ≤ 0.0001.

**Figure 2 nanomaterials-15-01478-f002:**
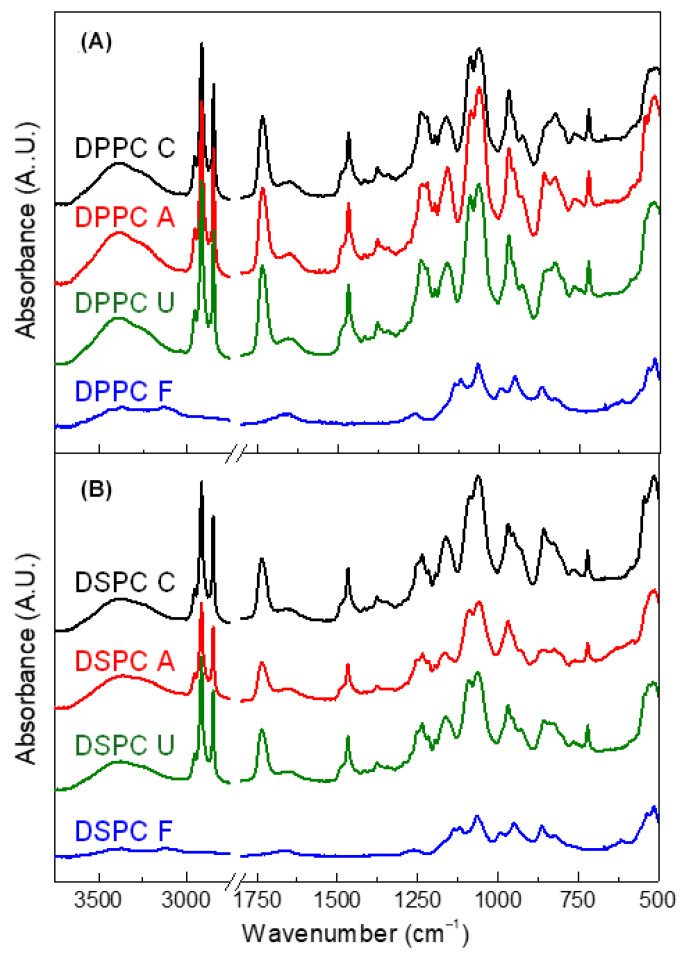
IR spectra of DPPC and DSPC liposomes after sterilization. (**A**) IR spectra of DPPC liposomes before and after sterilization by autoclaving, UV irradiation, and filtration. (**B**) IR spectra of DSPC liposomes before and after the same sterilization treatments.

**Figure 3 nanomaterials-15-01478-f003:**
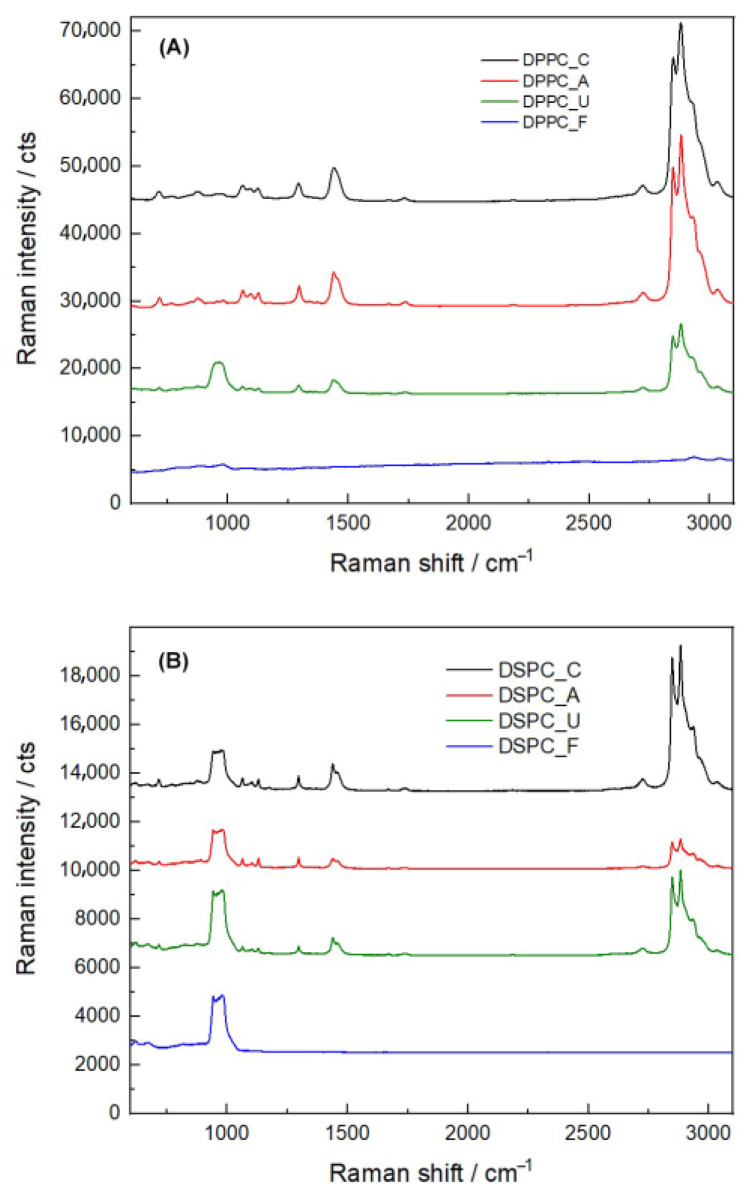
Raman spectral analysis of DPPC and DSPC liposomes before and after sterilization. (**A**) Raman spectra of DPPC liposomes highlighting changes in characteristic peaks after autoclaving, UV irradiation, and filtration. (**B**) Raman spectra of DSPC liposomes following the same sterilization treatments.

**Figure 4 nanomaterials-15-01478-f004:**
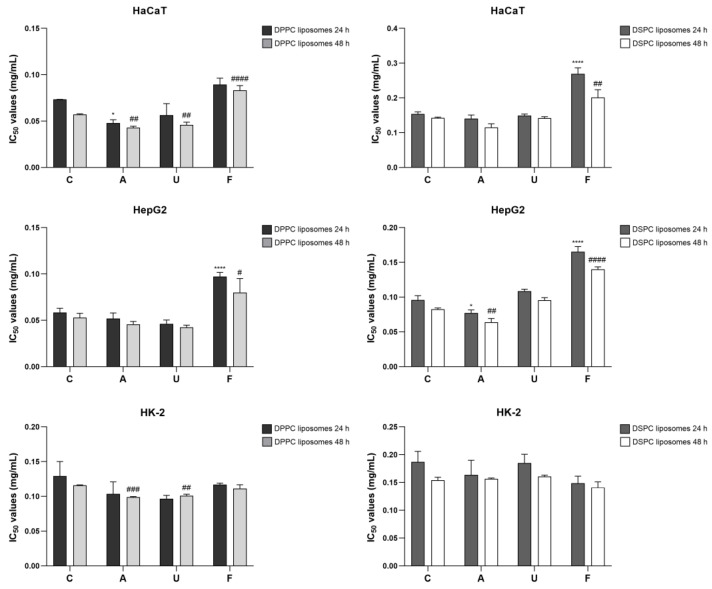
Cytotoxicity assessment of DPPC and DSPC liposomes after sterilization. The IC_50_ values of DPPC and DSPC liposomes in various cell lines (HaCaT, HepG2, and HK-2) after sterilization by autoclaving, UV irradiation, and filtration. Statistical significance: * *p* ≤ 0.05, **** *p* ≤ 0.0001 compared to the control at 24 h; # *p* ≤ 0.05, ## *p* ≤ 0.01, ### *p* ≤ 0.001, #### *p* ≤ 0.0001 compared to the control at 48 h.

**Table 1 nanomaterials-15-01478-t001:** Physicochemical characteristics of DPPC and DSPC liposomes before and after sterilization.

**Sterilization Treatments**	Z-Average (d.nm)	PDI	Zeta Potential (mV)
DPPC liposome			
Control	118.77 ± 0.65	0.19 ± 0.02	43.92 ± 2.20
Autoclaving	103.80 ± 1.68 ***	0.22 ± 0.01	44.17 ± 1.25
UV irradiation	117.83 ± 3.29	0.21 ± 0.02	39.58 ± 7.89
Filtration	114.33 ± 2.38	0.18 ± 0.02	28.21 ± 8.05 *
DSPC liposome			
Control	124.70 ± 1.82	0.21 ± 0.01	56.67 ± 4.97
Autoclaving	108.10 ± 0.30 ****	0.22 ± 0.03	57.98 ± 0.84
UV irradiation	128.10 ± 2.00	0.24 ± 0.02	58.82 ± 1.09
Filtration	114.00 ± 1.71 ***	0.17 ± 0.03	45.02 ± 0.40 **

Notes: Statistical significance: * *p* ≤ 0.05; ** *p* ≤ 0.01, *** *p* ≤ 0.001, **** *p* ≤ 0.0001 compared with control.

**Table 2 nanomaterials-15-01478-t002:** pH values of DPPC and DSPC liposomes before and after sterilization.

**Sterilization Treatments**	**pH**	
	**DPPC Liposomes**	**DSPC Liposome**
Control	4.22 ± 0.02	4.68 ± 0.03
Autoclaving	4.06 ± 0.01 ****	4.57 ± 0.03 **
UV irradiation	4.18 ± 0.01 *	4.68 ± 0.02
Filtration	4.30 ± 0.02 ***	4.52 ± 0.01 ***

Notes: Statistical significance: * *p* ≤ 0.05, ** *p* ≤ 0.01, *** *p* ≤ 0.001, and **** *p* ≤ 0.0001 compared with control.

**Table 3 nanomaterials-15-01478-t003:** IR Band assignments for DPPC and DSPC liposomes [25].

Wavenumber (cm^–1^)		Assignment
DPPC	DSPC	
721	721	CH_2_ rocking
824	827	P–O asymmetric stretching
969	969	CN asymmetric stretching, (CH_3_)_3_N^+^
1063	1062	C–O–P stretching, CO–O–C symmetric stretching
1089	1088	PO_2_^–^ symmetric stretching
1163	1162	CO–O–C asymmetric stretching
1242	1236	PO_2_^–^ asymmetric stretching
1377	1376	CH_3_ symmetric bending
1467	1467	CH_3_ asymmetric bending, CH_2_ scissoring
1734	1736	C=O stretching
2849	2849	CH_2_ symmetric stretching
2916	2915	CH_2_ asymmetric stretching
2955	2955	CH_3_ asymmetric stretching

**Table 4 nanomaterials-15-01478-t004:** Phospholipid content of DPPC and DSPC liposomes before and after sterilization.

**Sterilization Treatments**	**Phospholipid (mg/dL)**			
	**DPPC Liposomes**	**Percentage (%)**	**DSPC Liposomes**	**Percentage (%)**
Control	304.59 ± 7.47	100.00	251.36 ± 11.03	100.00
Autoclaving	268.89 ± 6.85 ***	88.30 ± 2.52	248.92 ± 4.13	99.11 ± 2.73
UV irradiation	279.83 ± 1.38 **	91.91 ± 2.10	241.86 ± 2.97	96.32 ± 3.61
Filtration	181.27 ± 5.89 ****	59.51 ± 1.04	58.70 ± 7.58 ****	23.39 ± 3.21

Notes: Statistical significance: ** *p* ≤ 0.01, *** *p* ≤ 0.001, and **** *p* ≤ 0.0001 compared with control.

## Data Availability

The raw data supporting the conclusions of this article will be made available by the authors on request.

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
