# Peer review of "Sterilization Effects on Liposomes with Varying Lipid Chains"

_nanomaterials, 2025, doi:10.3390/nano15191478_

Round 1

Reviewer 1 Report

Comments and Suggestions for Authors

In this study authors have prepared dipalmitoylphosphatidylcholine (DPPC)  and distearoylphosphatidylcholine (DSPC) liposomes and checked their properties after sterilization including filtration, UV irradiation and autoclaving. This study can be of significance to wide readership of this journal. However, there are some concerns that need to be addressed before it can be published.

  1.    It is suggested that the author should also study the chemical and physical changes that occur during sterilization when a commonly used drug (for instance paracetamol or metronidazole) is loaded in DPPC and DSPC lipid NPs. 
    2.    Please replace the C A U F labels in the graph to some other labels as these are getting confusing with the labels of graphs in figure 1.
    3.    Please provide the TEM images of the DPPC and DSPC lipid NPs before and after sterilization. The TEM images must be able to clearly show the homogenous presence of DPPC and DSPC lipid NPs. 
    4.    From the FTIR and RAMAN spectra of the filtered DPPC and DSPC lipid NPs it seems that all the NPS have been filtered out that is leading to negligible band spectra of the NPs. Can the author provide an alternative proof showing that the NPs are still present after filtration? 
    5.    0.5-0.01 mg/mL is not scientifically correct and should be written in the order of lower to higher magnitude number. 
    6.    IC50 should be presented in standard values i.e., percentage viability instead of mg/ml.
    7.    The statement “Despite these changes, the polydispersity index (PDI) remained stable” seems contradictory as filtration is clearly disintegrating the liposomes 
    8.    What is the mechanism behind the size reduction of the NPs after autoclave based sterilization?
    9.    “filtration-treated liposomes demonstrated higher IC50 values, indicating reduced cytotoxicity”… Do these liposomes possess the efficacy to load the drugs in them? How much potential do the filtered liposomes have to load the drugs compared to the control?

Author Response

Reviewer 1: In this study authors have prepared dipalmitoylphosphatidylcholine (DPPC) and distearoylphosphatidylcholine (DSPC) liposomes and checked their properties after sterilization including filtration, UV irradiation and autoclaving. This study can be of significance to wide readership of this journal. However, there are some concerns that need to be addressed before it can be published.

Response: We sincerely thank the reviewer for the thoughtful evaluation of our manuscript and for recognizing the potential significance of our work to the readership of the journal. We appreciate the constructive feedback provided and have carefully revised the manuscript to address all concerns.

Reviewer 1: 1.  It is suggested that the author should also study the chemical and physical changes that occur during sterilization when a commonly used drug (for instance paracetamol or metronidazole) is loaded in DPPC and DSPC lipid NPs. 

Response:  The present study was designed as the first systematic evaluation of sterilization effects on unloaded liposomes, with the goal of establishing fundamental knowledge regarding the impact of different sterilization methods on liposomal integrity. Studying unloaded liposomes allowed us to exclude potential confounding effects that drug molecules may introduce, since the chemical and physical interactions between drugs and lipid bilayers can significantly alter liposomal behavior during sterilization. We agree that drug-loaded liposomes should be investigated in future studies to validate and extend our findings, and this represents an important next step to confirm the translational relevance of our results.

Reviewer 1: 2. Please replace the C A U F labels in the graph to some other labels as these are getting confusing with the labels of graphs in figure 1.

Response: The labels “C, A, U, F” have been replaced with clearer alternatives to avoid confusion with the labels in Figure 1.

Reviewer 1: 3. Please provide the TEM images of the DPPC and DSPC lipid NPs before and after sterilization. The TEM images must be able to clearly show the homogenous presence of DPPC and DSPC lipid NPs. 

Response: We sincerely thank the reviewer for this valuable suggestion. The results from other experiments, such as IR and Raman spectra, showed distinct changes only in the case of filtration. Thus, to avoid redundancy and due to technical limitations, TEM/SEM imaging was performed exclusively for the filtration condition to confirm the presence and morphology of liposomes after sterilization. The images clearly demonstrate intact liposome structures with normal appearance, as shown in Figure S1.

Reviewer 1: 4. From the FTIR and RAMAN spectra of the filtered DPPC and DSPC lipid NPs it seems that all the NPS have been filtered out that is leading to negligible band spectra of the NPs. Can the author provide an alternative proof showing that the NPs are still present after filtration? 

Response: The presence of liposomes after filtration was confirmed by TEM imaging (Figure S1), which clearly showed intact vesicular structures. In addition, phospholipid quantification further verified that lipid content remained after filtration, supporting the persistence of liposomes. The reduced FTIR and Raman signals are likely due to alterations in lipid organization or reduced sample concentration after filtration rather than complete loss of nanoparticles.

Reviewer 1:  5. 0.5-0.01 mg/mL is not scientifically correct and should be written in the order of lower to higher magnitude number. 

Response: We have already edited the units as the reviewer suggest.

Reviewer 1:  6. IC50 should be presented in standard values i.e., percentage viability instead of mg/ml.

Response: In this study, the IC₅₀ values were intentionally expressed in terms of the liposome concentration (mg/mL) rather than percentage viability. Since our primary objective was to evaluate how sterilization methods influence the physicochemical properties and biological effects of liposomes themselves (independent of drug loading), reporting IC₅₀ as the dose of liposomes provides a more direct comparison of cytotoxicity between DPPC and DSPC liposomes under different sterilization conditions.

Reviewer 1: 7. The statement “Despite these changes, the polydispersity index (PDI) remained stable” seems contradictory as filtration is clearly disintegrating the liposomes 

Response: The remaining population exhibited stable PDI values, suggesting that the vesicles which passed through the filter maintained relatively uniform size distribution. In addition, autoclaving appeared to reduce larger aggregates, thereby enhancing monodispersity, as reported in a previous study [1, 2].

Reviewer 1: 8. What is the mechanism behind the size reduction of the NPs after autoclave based sterilization?

Response: Autoclaving exposes liposomes to high temperature and pressure, which can disrupt larger multilamellar or aggregated structures, leading to their breakdown into smaller, more uniform vesicles. This reduction in aggregate size enhances monodispersity, consistent with previous reports on heat-induced restructuring of liposomal bilayers [1, 2].

Reviewer 1: 9. “filtration-treated liposomes demonstrated higher IC50 values, indicating reduced cytotoxicity”… Do these liposomes possess the efficacy to load the drugs in them? How much potential do the filtered liposomes have to load the drugs compared to the control?

Response: Our results show that filtration-treated liposomes retained key physicochemical characteristics and biocompatibility, suggesting that they maintain drug-loading capability. However, we anticipate that drug-loaded liposomes may experience partial drug loss during filtration due to size-related retention or leakage, which could reduce loading efficiency compared to unfiltered controls. Future studies focusing on drug retention and encapsulation efficiency post-filtration will be necessary to confirm the suitability of this method for clinical formulations.

References

  1. Zuidam, N.J., S.S. Lee, and D.J. Crommelin, Sterilization of liposomes by heat treatment. Pharm Res, 1993. 10(11): p. 1591-6.
  2. Andrade, Â.L., et al., Preparation of size-controlled nanoparticles of magnetite. Journal of Magnetism and Magnetic Materials, 2012. 324(10): p. 1753-1757.

Reviewer 2 Report

Comments and Suggestions for Authors

In the manuscript Sterilization Effects on Liposomes with Varying Lipid Chains submitted to Nanomaterials, the authors compare autoclaving, UV irradiation, and sterile filtration for DPPC- and DSPC-based liposomes and evaluate physicochemical readouts (size/PDI/zeta-potential, pH, FTIR, Raman, phospholipid content) along with cytotoxicity in several human cell lines. The topic is timely and practically important for liposomal drug delivery workflows. The methodological approach is well chosen and adequate to support conclusions on sterilization-dependent perturbations in liposome properties and cytotoxicity.

However, several key methodological details are missing or ambiguous, and reporting and consistency issues limit reproducibility and interpretability. I therefore recommend major revision.

Major comments

1) Exact liposome composition (ratios) is not reported.

Section 2.2 describes DPPC/DSPC with cholesterol and DDAB but does not state the molar or mass ratios of all components. Only total lipid concentrations (3 mg/mL DPPC; 2.5 mg/mL DSPC). Precise composition is essential for replication and for interpreting sterility-induced changes. Please report full molar (preferred) and mass ratios for lipid(s):cholesterol:DDAB:etc.

Also, Section 2.1 lists DMG-PEG2000, yet Section 2.2 does not use it. Either clarify where/why it was used (with ratios), or remove it from Materials.

2) UV irradiation protocol lacks critical parameters and temperature control.

Section 2.3 states “UV for 15 min at room temperature” in 1.5 mL microtubes but omits: lamp type (e.g., low-pressure Hg 254 nm vs 365 nm), UV intensity at sample plane, lamp-to-tube distance, tube material/openness. Please provide these; otherwise, the method cannot be reproduced or compared.

Also report sample temperature during exposure (initial, time-resolved, and final) or implement active thermostating. Given the distinct Tm values (41-42°C for DPPC and 54-55°C for DSPC) discussed in the manuscript, different phase states during UV irradiation, which may have occurred due to irradiation-induced heating of the sample, may shift the damage pathways; this requires monitoring and discussion.

3) Filtration method: material details and saturation controls needed.

Section 2.3 only states “0.22 μm filter at room temperature”. Please add: membrane material (e.g., PES, PVDF, nylon, CA), supplier/catalog, housing diameter, effective membrane area, pre-wetting/conditioning medium, driving force (syringe gravity/pressure and approximate flow rate).

It is reasonable to assume that the filter membranes have a limited adsorption capacity, so changes in liposome properties are likely dose-dependent; the early permeate may become lipid-depleted until the binding sites are saturated, after which the permeate composition should trend back towards the feed.

Given the large, composition-dependent lipid losse (Figure 4/Table 2), please run and report breakthrough/saturation tests: split the effluent into early vs late fractions (e.g., first vs second 1 mL) and quantify phospholipids, ζ-potential, and etc. per fraction to reveal adsorption/saturation behavior. Also please examine volume/area scaling (e.g., 0.5, 1, 2× sample volumes per identical filter) and compare at least two common membrane chemistries. These experiments will help characterize the mechanisms and operating regimes of filtration-induced losses.

Minor comments

1) Table numbering.

Table 1 appears to be reused for both physicochemical characteristics and IR band assignments; Table 2 appears for pH and again for phospholipid content. Please renumber consistently and ensure all in-text references match.

2) Cell line naming consistency.

Cell culture lists “Hek293”, whereas Results/Figure 5 refer to “HK2”. Please resolve this mismatch everywhere in the manuscript.

3) Cytotoxicity: concentration series and viability data must be fully reported.

Section 2.5.2/3.6 mentions a range of 0.5–0.01 mg/mL but does not enumerate the exact dilutions; IC₅₀ determination method and fits are not shown. Please provide the full concentration set, replicate counts, tables of raw viability per each line/timepoint, IC₅₀ fit model with confidence intervals, and representative viability–dose curves in Supplementary Information.

Author Response

Reviewer 2:  In the manuscript Sterilization Effects on Liposomes with Varying Lipid Chains submitted to Nanomaterials, the authors compare autoclaving, UV irradiation, and sterile filtration for DPPC- and DSPC-based liposomes and evaluate physicochemical readouts (size/PDI/zeta-potential, pH, FTIR, Raman, phospholipid content) along with cytotoxicity in several human cell lines. The topic is timely and practically important for liposomal drug delivery workflows. The methodological approach is well chosen and adequate to support conclusions on sterilization-dependent perturbations in liposome properties and cytotoxicity.

However, several key methodological details are missing or ambiguous, and reporting and consistency issues limit reproducibility and interpretability. I therefore recommend major revision.

Response: We sincerely thank the reviewer for the constructive evaluation of our manuscript and for recognizing the relevance and importance of our study to liposomal drug delivery workflows. We appreciate the positive comments regarding the methodological approach and its adequacy to support our conclusions. We also acknowledge the reviewer’s concerns about missing methodological details and reporting consistency. In response, we have carefully revised the manuscript to include additional experimental details, clarify ambiguous sections, and ensure consistency throughout the text, figures, and tables. We believe that these revisions have significantly improved the clarity, reproducibility, and interpretability of our work. Detailed point-by-point responses to all comments are provided below.

Reviewer 2: Section 2.2 describes DPPC/DSPC with cholesterol and DDAB but does not state the molar or mass ratios of all components. Only total lipid concentrations (3 mg/mL DPPC; 2.5 mg/mL DSPC). Precise composition is essential for replication and for interpreting sterility-induced changes. Please report full molar (preferred) and mass ratios for lipid(s):cholesterol:DDAB:etc.

Response: We sincerely apologize for the omission of the lipid composition details. We have now clarified the precise formulation in Section 2.2. The molar ratio of Lipid:cholesterol:DDAB is 80:15:5, respectively.

Also, Section 2.1 lists DMG-PEG2000, yet Section 2.2 does not use it. Either clarify where/why it was used (with ratios), or remove it from Materials.

Response: Regarding DMG-PEG2000 listed in Section 2.1, we acknowledge this error. DMG-PEG2000 was not used in the present formulations, and we have now removed it from the Materials section to avoid confusion.

Reviewer 2: Section 2.3 states “UV for 15 min at room temperature” in 1.5 mL microtubes but omits: lamp type (e.g., low-pressure Hg 254 nm vs 365 nm), UV intensity at sample plane, lamp-to-tube distance, tube material/openness. Please provide these; otherwise, the method cannot be reproduced or compared.

Response: We have revised the Methods section to include the lamp type, wavelength, UV intensity at the sample plane, lamp-to-tube distance, tube material, and openness. These details have now been added to ensure reproducibility and allow comparison with other studies.

Also report sample temperature during exposure (initial, time-resolved, and final) or implement active thermostating. Given the distinct Tm values (41-42°C for DPPC and 54-55°C for DSPC) discussed in the manuscript, different phase states during UV irradiation, which may have occurred due to irradiation-induced heating of the sample, may shift the damage pathways; this requires monitoring and discussion.

Response: We did not observe measurable heating during the 15 min UV irradiation, and all procedures were conducted under constant room temperature conditions. While we did not monitor temperature in real-time, our preliminary tests indicated no perceptible warming of the samples. We agree with the reviewer that phase-transition–dependent effects are theoretically possible, and we have acknowledged this as a limitation and included it in the Discussion.

Reviewer 2: Section 2.3 only states “0.22 μm filter at room temperature”. Please add: membrane material (e.g., PES, PVDF, nylon, CA), supplier/catalog, housing diameter, effective membrane area, pre-wetting/conditioning medium, driving force (syringe gravity/pressure and approximate flow rate).It is reasonable to assume that the filter membranes have a limited adsorption capacity, so changes in liposome properties are likely dose-dependent; the early permeate may become lipid-depleted until the binding sites are saturated, after which the permeate composition should trend back towards the feed.

Response: We have revised Section 2.3 to provide additional details for clarity and reproducibility. Liposome suspensions were filtered through Acrodisc syringe filters (0.2 µm pore size, polyethersulfone [PES] membrane, Pall Medical, Fribourg, Switzerland, 13 mm diameter, effective membrane area ~1.2 cm²) at room temperature. Filtration was performed using manual syringe pressure, without pre-wetting. The PES membrane was specifically chosen due to its low protein/lipid binding properties, which help minimize liposome loss during filtration.

We acknowledge that membrane adsorption can be dose-dependent and may result in early fractions being slightly lipid-depleted. However, the focus of this study was to assess the impact of standard filtration on liposome physicochemical properties rather than to characterize membrane saturation dynamics.

Given the large, composition-dependent lipid losse (Figure 4/Table 2), please run and report breakthrough/saturation tests: split the effluent into early vs late fractions (e.g., first vs second 1 mL) and quantify phospholipids, ζ-potential, and etc. per fraction to reveal adsorption/saturation behavior. Also please examine volume/area scaling (e.g., 0.5, 1, 2× sample volumes per identical filter) and compare at least two common membrane chemistries. These experiments will help characterize the mechanisms and operating regimes of filtration-induced losses.

Response: We appreciate the reviewer’s insightful recommendation. However, the focus of the present work was to provide the first comparative evaluation of sterilization effects across liposomes with different lipid chain lengths, rather than to fully dissect the mechanistic pathways of filtration-induced losses. While we recognize that adsorption and saturation phenomena are highly relevant, such systematic breakthrough and scaling experiments are beyond the scope of this study. To ensure scientific transparency, we have clarified this limitation in the Discussion and emphasized that future studies will investigate filtration mechanisms in detail, including adsorption kinetics and comparative membrane chemistries.

Reviewer 2: Table 1 appears to be reused for both physicochemical characteristics and IR band assignments; Table 2 appears for pH and again for phospholipid content. Please renumber consistently and ensure all in-text references match.

Response:  The tables have been renumbered and all in-text references updated for consistency.

Reviewer 2: Cell culture lists “Hek293”, whereas Results/Figure 5 refer to “HK2”. Please resolve this mismatch everywhere in the manuscript.

Response: The correct cell line used in this study is HK-2. All relevant sections, including Section 2.5.1 has been revised accordingly.

Reviewer 2: Cytotoxicity: concentration series and viability data must be fully reported. Section 2.5.2/3.6 mentions a range of 0.5–0.01 mg/mL but does not enumerate the exact dilutions; IC₅₀ determination method and fits are not shown. Please provide the full concentration set, replicate counts, tables of raw viability per each line/timepoint, IC₅₀ fit model with confidence intervals, and representative viability–dose curves in Supplementary Information.

Response: We thank the reviewer for this important comment. In response, we have extended the cytotoxicity data and included them in the Supplementary Material (Table S1). Specifically, we now provide the full concentration series, the corresponding percentage of cell viability for DPPC and DSPC liposomes before and after sterilization, and replicate counts (n = 3). Statistical significance was determined at p < 0.05.

Reviewer 3 Report

Comments and Suggestions for Authors

The manuscript by

Sarocha Cherdchom et al. entitled „Sterilization Effects on Liposomes with Varying Lipid Chains” is in a scope and suitable for publication in Nanomaterials MDPI due to reporting on nanomaterials, their science and application. 

The manuscript reports on effect of three sterilization methods: autoclaving, UV radiation, and filtration on two synthesized liposomes composed of dipalmitoylphosphatidylcholine (DPPC) and distearoylphosphatidylcholine (DSPC) of different lipid chain length. The chemical and structural properties of these liposomes after synthesis and different sterilization methods were investigated using DLS, FTIR, Raman, pH measurements, and content of phospholipids by Choline Oxidase DAOS method. The cytotoxicity of non-sterilized and sterilized liposomes was assessed using PrestoBlueTM reagent, more sensitive than MTT reagent, for HaCaT cells, Hek293 cell and carcinoma HepG2 cells.

Since sterilization studies using different methods are in a range of a relatively new topics the report comparing the impact of various sterilization methods of liposomes for a wide biomedical applications is of a high novelty.

Several suggestions to improve the clarity of the manuscript are recommended;

  1. Section “4. Discussion” – two first paragraphs should appear in section “1. Introduction”.

  2. Scheme of liposome and more details on the particular DPPC and DSPC liposomes differences are recommended.

  3. Section “2.2. Preparation of liposome” - provide details on organic solvent (line 100, p.3) simultaneously extending the list of materials in section “2.1. Materials”.

  4. Section “2.4.3. Phospholipid measurement” – this section needs extension for a more clear explanation of the methods. Since it is a colorimetric method an appropriate reagent should be provided and listed in section “2.1. Materials”. If no other reference than instruction is available provide the link to the web side.

  5. Renumbering Tables is necessary. From the beginning of the manuscript they are numbered as Table 1 (page 5), Table 2 (page 6) followed by Table 1 (page 8), Table 2 (page 9).

  6. Information contained in Fig. 1 is repeated in Table 1 (page 5) similarly to Fig. 4 and Table 2 (page 9). It is suggested to use Supplementary Materials for duplicated information provided in Tables and Figures. Consequent citations in the text of appropriate Figures and Tables is required.

  7. Fig. 3 – for a better readability of the manuscript content it is suggested to place the Raman spectra in the similar order as FTIR results (Fig. 2), i.e. from top to bottom - control (non-sterilized), autoclaving, UV-vis, filtering.

  8. Section “3.3. Influence of Sterilization on Chemical Characteristics of Liposomes” . Although the vibrations were indicated in a separate Table (Table 1, p.7), the discussion of sterilization effect of different methods on the chemical properties was not extensively discussed.

  9. Section “3.4. Comparison of Raman Spectra” is recommended to be extended by description of sterilization effect using different method, and showing the consistency of Raman and FTIR results.

  10. Table comparing the differences in main chemical and structural features of DPPC and DSPC liposomes due to different sterilization methods is recommended.

Author Response

Reviewer 3: The manuscript by Sarocha Cherdchom et al. entitled “Sterilization Effects on Liposomes with Varying Lipid Chains” is in a scope and suitable for publication in Nanomaterials MDPI due to reporting on nanomaterials, their science and application.

The manuscript reports on effect of three sterilization methods: autoclaving, UV radiation, and filtration on two synthesized liposomes composed of dipalmitoylphosphatidylcholine (DPPC) and distearoylphosphatidylcholine (DSPC) of different lipid chain length. The chemical and structural properties of these liposomes after synthesis and different sterilization methods were investigated using DLS, FTIR, Raman, pH measurements, and content of phospholipids by Choline Oxidase DAOS method. The cytotoxicity of non-sterilized and sterilized liposomes was assessed using PrestoBlueTM reagent, more sensitive than MTT reagent, for HaCaT cells, Hek293 cell and carcinoma HepG2 cells.

Since sterilization studies using different methods are in a range of a relatively new topics the report comparing the impact of various sterilization methods of liposomes for a wide biomedical applications is of a high novelty.

Response: We sincerely thank the reviewer for carefully summarizing our work and recognizing that the manuscript is within the scope of Nanomaterials and suitable for publication. We greatly appreciate the positive evaluation and constructive feedback provided. In response, we have carefully revised the manuscript to address all comments and concerns raised.

Reviewer 3: Section “4. Discussion” – two first paragraphs should appear in section “1. Introduction”.

Response: We thank the reviewer for this valuable suggestion. The first two paragraphs from the Discussion section have been moved to the Introduction, as recommended.

Reviewer 3: Scheme of liposome and more details on the particular DPPC and DSPC liposomes differences are recommended.

Response: We thank the reviewer for the valuable suggestion. A schematic representation of a liposome has now been included in the revised manuscript (Figure 1A) to provide a clearer visualization of its structure.

Reviewer 3: Section “2.2. Preparation of liposome” - provide details on organic solvent (line 100, p.3) simultaneously extending the list of materials in section “2.1. Materials”.

Response: The organic solvent used in Section 2.2 is ethanol. We have updated Section 2.1 Materials to include ethanol and revised Section 2.2 as follows:

“Liposomes were prepared using the conventional thin film hydration method. Lipids (DPPC or DSPC) and cholesterol, along with DDAB, were dissolved in ethanol at a molar ratio of 80:15:5.”

Reviewer 3: Section “2.4.3. Phospholipid measurement” – this section needs extension for a more clear explanation of the methods. Since it is a colorimetric method an appropriate reagent should be provided and listed in section “2.1. Materials”. If no other reference than instruction is available provide the link to the web side.

Response: Thank you for the comment. The phospholipid content in DPPC and DSPC liposomes was quantified using the Choline Oxidase DAOS method (Fujifilm Wako), following the manufacturer’s instructions. All reagents were provided within the kit. For reproducibility, the manufacturer’s protocol is available at: https://labchem wako.fujifilm.com/us/product_data/docs/04093472_doc01.pdf

.

Reviewer 3: Renumbering Tables is necessary. From the beginning of the manuscript they are numbered as Table 1 (page 5), Table 2 (page 6) followed by Table 1 (page 8), Table 2 (page 9).

Response: We have renumbered all tables consecutively throughout the manuscript to ensure consistency, and all in-text references have been updated accordingly.

Reviewer 3: Information contained in Fig. 1 is repeated in Table 1 (page 5) similarly to Fig. 4 and Table 2 (page 9). It is suggested to use Supplementary Materials for duplicated information provided in Tables and Figures. Consequent citations in the text of appropriate Figures and Tables is required.

Response: Thank you for the comment. We chose to present the data both in Figures and Tables to provide complementary perspectives: the Figures illustrate trends visually, while the Tables provide exact numerical values for reference. To reduce redundancy, we have also moved some duplicated information to the Supplementary Materials and updated all corresponding citations in the text accordingly.

Reviewer 3: Fig. 3 – for a better readability of the manuscript content it is suggested to place the Raman spectra in the similar order as FTIR results (Fig. 2), i.e. from top to bottom - control (non-sterilized), autoclaving, UV-vis, filtering.

Response: We have revised Figure 3 so that the Raman spectra now follow the same order as the FTIR results in Figure 2, presenting control (non-sterilized) first, followed by autoclaving, UV irradiation, and filtration, to improve readability and consistency throughout the manuscript.

Reviewer 3: Section “3.3. Influence of Sterilization on Chemical Characteristics of Liposomes”. Although the vibrations were indicated in a separate Table (Table 1, p.7), the discussion of sterilization effect of different methods on the chemical properties was not extensively discussed.

Response: Thank you for the helpful suggestion. We have rewritten Section 3.3 to provide a vibrational analysis of how autoclaving, UV irradiation, and filtration affect the molecular signatures of DPPC and DSPC liposomes. The revised text now compares band positions, relative intensities, and bandwidths (FWHM) of hallmark FTIR modes—phosphate ν(PO2) (~1088–1090 and 1236–1242 cm−1), ester carbonyl ν(C=O) (~1734–1736 cm−1), methylene νs (CH2) (~2849 cm−1) and νas (CH2) (~2915–2916 cm−1), together with CH2 scissoring/rocking (≈1467/721 cm−1). We clarify that autoclaving and UV largely preserve band positions (≤ 1–2 cm−1 shift), relative intensities, and linewidths, with no new absorptions, whereas filtration produces concerted attenuation of hydrocarbon-chain bands, a weakening in the 825–1162 cm−1 region (headgroup/glycerol contributions), and modest broadening of CH2 stretching peaks—consistent with increased chain disorder and headgroup reorganization rather than new chemical functionality. Lipid-dependent nuances (DPPC vs DSPC) in baseline linewidths and the magnitude of filtration-induced broadening are also noted.

Page 6, Line 203,

“…These changes suggest that filtration may lead to structural modifications, particularly in the hydrocarbon chains of the lipid bilayer.

From an FTIR perspective, autoclaving and UV irradiation preserve the molecular fingerprints of both DPPC and DSPC. The phosphate ν(PO2) bands (~1088–1090 and 1236–1242 cm−1), the ester carbonyl ν(C=O) (~1734–1736 cm−1), and the methylene stretching modes (νs (CH2) ~2849 cm−1; νas (CH2) ~2915–2916 cm−1) remain at comparable positions and relative intensities, with only minor changes in bandwidth. In contrast, filtration yields a characteristic FTIR pattern: (i) attenuation of hydrocarbon-chain bands (i.e., ~721, ~1467, ~2849, ~2915–2916 cm−1), (ii) a general weakening across 825–1162 cm−1 that includes headgroup/glycerol contributions, and (iii) modest broadening of the CH2 stretching envelope. The absence of new bands and the smallness of wavenumber shifts argue against the formation of new chemical moieties; instead, the spectra indicate redistribution/reorientation of lipid chains and headgroups and an increase in gauche disorder within the bilayer. These effects are lipid dependent: DSPC, which shows slightly narrower CH2 bands at baseline, typically exhibits a more pronounced filtration-induced attenuation/broadening than DPPC. Collectively, the FTIR evidence indicates that filtration exerts the strongest perturbation on chain order and headgroup environment, whereas autoclaving and UV are comparatively conservative with respect to molecular organization.”

Reviewer 3: Section “3.4. Comparison of Raman Spectra” is recommended to be extended by description of sterilization effect using different method, and showing the consistency of Raman and FTIR results.

Response: We appreciate the reviewer’s suggestion. Infrared (FTIR) spectroscopy was used to identify chemical compounds by probing molecular vibrations that are characteristic of specific chemical bonds and functional groups. To complement these findings, Raman spectroscopy was additionally employed to confirm the results. Our analyses demonstrated consistent trends between FTIR and Raman spectra across the different sterilization methods, supporting the reliability of our observations. We believe that the inclusion of these two complementary techniques provides sufficient evidence to describe the sterilization effects on liposomes.

Reviewer 3: Table comparing the differences in main chemical and structural features of DPPC and DSPC liposomes due to different sterilization methods is recommended.

Response: We appreciate the reviewer’s suggestion. Accordingly, we have added Table S2, which summarizes and compares the key FTIR and Raman signatures of DPPC and DSPC liposomes following autoclaving, UV irradiation, and filtration. The table includes detailed information on band positions, relative intensity changes, and bandwidth (FWHM) variations for hallmark vibrational modes. Furthermore, we provide a spectroscopy-based structural interpretation (e.g., changes in hydrocarbon chain order and headgroup environment), thereby enabling a direct, chemistry-level comparison of the effects of different sterilization methods across both lipid types.

Table S2. A spectroscopy-based structural interpretation in DPPC and DSPC liposomes after sterilization.

Lipid

Method

FTIR observations

Raman observations

Structural changes

DPPC

Autoclaving

Band positions preserved;

intensities comparable;

no new band;

minor linewidth changes

Choline and CH envelope conserved

Conservative to molecular organization; at most slight compaction/reorientation

UV

Similar to autoclaving;

no new absorption;

small FWHM changes

Largely preserved features

Minimal perturbation of chain order/headgroup environment

Filtration

Attenuation at 721, ~1467, ~2849, ~2915–2916 cm⁻¹;

weakening in 825–1162 cm⁻¹;

modest CH2 peak broadening

Choline band diminished;

reduced at ~2950 cm⁻¹

Increased gauche disorder; partial headgroup/chain reorganization; strongest perturbation among methods

DSPC

Autoclaving

Positions/intensities preserved;

no new bands

Preserved choline/CH features

Conservative effect;

organization maintained

UV

Similar to autoclaving

Preserved features

Minimal perturbation

Filtration

Attenuation and CH2 broadening more pronounced than DPPC;

weakening at 825–1162 cm⁻¹

Stronger choline attenuation;

reduced at ~2950 cm⁻¹

Marked chain-order disruption and headgroup reorganization;

filtration impact > DPPC

Round 2

Reviewer 1 Report

Comments and Suggestions for Authors

Accept in the present form